# Ecoepidemiology and Potential Transmission of *Vibrio cholerae* among Different Environmental Niches: An Upcoming Threat in Egypt

**DOI:** 10.3390/pathogens10020190

**Published:** 2021-02-10

**Authors:** Eman M. Ismail, Mona Kadry, Esraa A. Elshafiee, Eman Ragab, Eman A. Morsy, Omar Rizk, Manal M. Zaki

**Affiliations:** 1Department of Veterinary Hygiene and Management, Faculty of Veterinary Medicine, Cairo University, Giza 12211, Egypt; drmanalmoustafa2008@cu.edu.eg; 2Department of Zoonoses, Faculty of Veterinary Medicine, Cairo University, Giza 12211, Egypt; Mona.kadry.mohamed@cu.edu.eg (M.K.); esraa_elshafiee@cu.edu.eg (E.A.E.); 3Department of Microbiology, Faculty of Veterinary Medicine, Cairo University, Giza 12211, Egypt; eman_ragab2008@cu.edu.eg; 4Department of Poultry Diseases, Faculty of Veterinary Medicine, Cairo University, Giza 12211, Egypt; emananter@cu.edu.eg; 5Faculty of Biotechnology—MSA University, 6th October City 12573, Egypt; Omar.mohamed91@msa.edu.eg

**Keywords:** cholera, ecoepidemiology, viable but not culturable (VBNC), latent endemicity, Nile River, host interactions, waterfowls, good managemental practices, *Omp W* gene

## Abstract

Cholera is a negative public health event caused by *Vibrio cholerae.* Although *V. cholerae* is abundant in natural environments, its pattern and transmission between different niches remain puzzling and interrelated. Our study aimed to investigate the occurrence of nonpathogenic *V. cholerae* in the natural environment during endemicity periods. It also aimed to highlight the role of molecular ecoepidemiology in mapping the routes of spread, transmission, and prevention of possible future cholera outbreaks. *V. cholerae* was detected in different aquatic environments, waterfowl, and poultry farms located along the length of the Nile River in Giza, Cairo, and Delta provinces, Egypt. After polymerase chain reaction amplification of the specific target outer membrane gene *(Omp W)* of suspected isolates, we performed sequence analysis, eventually using phylogenetic tree analysis to illustrate the possible epidemiological relationships between different sequences. Data revealed a significant variation in the physicochemical conditions of the examined Nile districts related to temporal, spatial, and anthropogenic activities. Moreover, data showed an evident association between *V. cholerae* and the clinically diseased *Synodontis schall* fish. We found that the environmental distress triggered by the salinity shift and elevated temperature in the Middle Delta of the Nile River affects the pathogenesis of *V. cholerae*, in addition to the characteristics of fish host inhabiting the Rosetta Branch at Kafr El-Zayat, El-Gharbia province, Egypt. In addition, we noted a significant relationship between *V. cholerae* and poultry sources that feed on the Nile dikes close to the examined districts. Sequence analysis revealed clustering of the waterfowl and broiler chicken isolates with human and aquatic isolated sequences retrieved from the GenBank databases. From the obtained data, we hypothesized that waterfowl act as a potential vector for the intermediate transmission of cholera. Therefore, continuous monitoring of Nile water quality and mitigation of Nile River pollution, in addition to following good managemental practices (GMPs), general hygienic guidelines, and biosecurity in the field of animal production and industry, might be the way to break this cyclic transmission between human, aquatic, and animal sectors.

## 1. Introduction

Recently, there has been a massive increase in data published regarding public health events that emphasize the role of epidemiology and the environment in the transmission cycle of many threatened zoonotic diseases. Cholera is an upsetting water- and food-borne epidemic and a pandemic life-threatening diarrheal disease caused by *Vibrio cholerae* that affects different hosts and species [1,2]. The gram-negative bacterium *V. cholerae* is a facultative pathogen that has both human and environmental stages in its life cycle [3].

*V. cholerae* has the unique capability to persist in two forms: viable but not culturable (VBNC) and conditionally viable environmental cells (CVEC). The VBNC is the latent endemic form of *V. cholerae* that occurs in the natural environment in response to temporal, seasonal, and geographical variations and/or nutrient deprivation [2,4,5]. This state of endemicity has been hypothesized to be based on four theories: animal reservoirs theory (nonhuman persistence), chronicity theory (human persistence), human continuous transmission theory, and environmental persistence (aquatic environment as a reservoir) [6,7,8,9,10]. Until 1970, *V. cholerae* was thought to have a limited ability to persist and survive in the environment. However, insufficient data regarding the first three theories, along with significant evidence of several epidemiological, environmental, and laboratory studies, challenged this view [4,9,11]. Since then, researchers have claimed that the aquatic environment is the natural habitat of cholerae, where several factors affect their persistence, growth, and survival in the natural ecosystem.

In Egypt, the Nile River is considered the principal fresh aquatic resource and is referred to as “the artery of life”. It covers most of the demands for drinking water, irrigation, and industry [12]. Moreover, it is the most important source of many freshwater and crustacean food organisms. Environmentally, the high self-purification and assimilation capacity of water from the Nile has resulted in the Nile being listed as a moderately clean river. Even so, it suffers from localized pollution challenges that affect the water quality and ecosystem along its length from Aswan to the Delta [13,14].

Epidemiologically, the role cholera toxin plays in the environment remains unclear. However, it is known that *V. cholerae* bacteria are commonly related to the chitin that contains zooplankton, especially copepods and chironomids [15,16]. They interact with several inhabitants to shape their potential virulence and pathogenicity. Thus, a clear distinction is essential between the cholera reservoirs and vectors and those that transmit *V. cholerae* only through mechanical attachment to their surfaces and tissues. Moreover, significant supporting data have reported that fish and waterfowl can directly transmit *V. cholerae* among different niches [4].

Interestingly, the transmission of *V. cholerae* is a complex interactive cycle among different environment, agent, and host characteristics and is associated with socioeconomic issues. On the other hand, faulty management practices during the animal production cycle can cause the transmission of many disease agents between different hosts, which promotes environmental persistence and the production of resistant modified microbial strains [17,18,19,20]. Furthermore, the low level of hygienic biosecurity measures in poultry farms is a threatening issue, as different pathogenic microorganisms have been identified in this setting, including cholera [21]. Hence, the improper management and disposal of poultry excreta could be linked to an epidemic outbreak of cholera.

Therefore, the aim of this study was to investigate the occurrence of nonpathogenic *V. cholerae* in the natural environment during the period of endemicity and to highlight the role of the molecular ecoepidemiology of *V. cholerae* to better understand the transmission cycle and host interactions among different environmental sources of aquatic organisms, waterfowl, poultry, and humans. Thus, breaking down the transmission cycle among the different niches can possibly help prevent cholera outbreaks.

## 2. Results

### 2.1. Phenotypic and Genotypic Identification of V. cholerae in Different Environmental Niches

We compared the occurrence of cholerae among different ecological sources. The data revealed that *V. cholerae* was abundant in the examined aquatic and poultry sources. In addition, a cyclic transmission between these different environmental niches was evident. On the other hand, *V. cholerae* was isolated and identified from 42% of examined water and sediment samples from different Nile environments, mostly from sediments (Table 1).

All suspected vibrio isolates collected from both aquatic (sediment, water, tilapia, shell fish, and seafood samples), besides poultry species (chickens, turkey, and waterfowls), were purified and identified biochemically at the species level (Appendix A). All suspected isolates were able to grow at 42 °C after 6 h of incubation with yellow-colored colonies on thiosulfate citrate bile salt agar plates. Moreover, the isolates tested positive for oxidase, ornithine decarboxylase, and gelatinase activity, whereas they tested negative for arginine hydrolysis and urease activity. Regarding salt tolerance, all isolates were able to grow in the absence of NaCl (0% and 3% NaCl).

We conducted polymerase chain reaction (PCR) for the genotypic detection of universal 16S rRNA gene in addition to the specific target outer membrane protein (*Omp W*) of *V. cholerae.* Data revealed that the suspected isolates from the examined sediments, water, finfish, crustacean seafood, domesticated poultry, and waterfowl belonged to the species of *V. cholerae* (Table 1).

### 2.2. Ecoepidemiology of V. cholerae in Aquatic Nile Environment (Water and Sediments)

Monitoring the physicochemical water conditions in different Nile environments revealed variations in water temperature, salinity, and pH in the examined districts along the Nile River from Giza to Delta. As predicted, the water temperature of the Nile varied greatly among the seasons. Alternatively, the variability of salinity and pH was affected by the geographic location of the different Nile districts.

As shown in Figure 1, the highest temperature was recorded at sites 3 and 5. In addition, the salinity of the Nile water differed along its length in Giza, Cairo, and El-Delta provinces. These variations in salinity categorize the Nile water spatially into three different salinity zones: fresh (0.5–2 ppt), brackish (2–15 ppt), and marine (>15 ppt). At site 3, there was a salinity shift of Nile water from the fresh to the brackish side, and the elevation continued in a seaward direction to the Mediterranean Sea Nile outlets, where the highest values of salinity were recorded at sites 4, 5, and 6, along the coastal line of the sea. The pH range was toward the alkaline side (7.56–8.5), with slight local differences but no distinct temporal variations observed.

#### 2.2.1. Occurrence of Cholerae and Spatial Variation (Salinity)

The occurrence of *V. cholerae* in marine and brackish water was higher than in freshwater, with a significant association of *V. cholerae* infection in the examined finfish samples collected from the brackish water environment at site 3, χ^2^ (2, N = 185) = 29.02, *p* < 0.0001 (Table 2).

#### 2.2.2. Occurrence of Cholerae Regarding Temporal Variation

With regard to temporal variation (Figure 2), the occurrence of *V. cholerae* in water and sediment samples during hot temperature was higher than in other zones. No cholerae were detected in examined water samples collected from either site 2 or 6. Meanwhile, there was a significant noticeable association between temperature increase and the occurrence of *V. cholerae* in the examined finfish and seafood crustacean samples, *χ*^2^(2, N = 185) = 5.95, *p* = 0.05.

### 2.3. Ecoepidemiology of V. cholerae and Aquatic Host Interactions

Regarding the ecoepidemiology of *V. cholerae* and aquatic animal reservoirs, we noted a different pattern of cholerae occurrence among the examined aquatic animals. Clinical examination of freshly caught finfish and crustacean seafood samples revealed that all were apparently healthy, except for *Synodontis schall* fish samples. Fish samples collected from site 3 had evident clinical signs related to septicemic bacterial infection (Figure 3a,b, arrows). These symptoms included ascites, eye opacity, edema and discoloration of the gills, and skin lesions. Postmortem examination of the fish species revealed signs of swelling of the internal organs, liver, spleen, and kidney. The fish were found to have abdominal dropsy.

Table 3 illustrates the significant correlation between the occurrence of *V. cholerae* and examined aquatic sources (cultured tilapia, catfish, and crustaceans) that inhabit different ecosystems, *χ*^2^ (2, N = 185) = 29.02, *p* < 0.0001. However, the occurrence of *V. cholerae* in shield head fish (*Synodontis schall*, 46/64, 72.0%) was significantly higher than in Nile tilapia (*Oreochromis niloticus*, 12/36, 33.3), *χ*^2^ (1, N = 100) = 14.05, *p* < 0.0001.

### 2.4. Occurrence of V. cholerae in Poultry Species and Waterfowls

Data in Table 4 reveal a significant relationship between the occurrence of *V. cholerae* and poultry sources collected from broiler, duck, and turkey farm feed on the Nile River near the examined aquatic Nile ecosystems from Giza to the Northern Delta provinces, *χ*^2^ (2, N = 115) = 13.66, *p* = 0.001.

Sequence analysis and comparing the sequences of the *Omp* W gene revealed 100% homology between waterfowl (duck) and broiler chicken isolates (Figure 4). Both sequences clustered with human, wastewater, and surface water isolated sequences retrieved from GenBank databases.

## 3. Discussion

Significant evidence of environmental, epidemiological, and laboratory findings of cholera investigations have proven that the aquatic environment is considered a natural reservoir of *V. cholerae.* This environmental theory makes the epidemiology of cholerae unclear, complex, and puzzling.

The epidemiology of *V. cholerae* relies on multifactorial environmental and social actions. Thus, organic fertilization of aquatic pond farms with animal and poultry wastes, sewage, and municipal wastewater could increase the possibility of *V. cholera* transmission to inhabited fish and crustaceans (environmental reservoirs). However, in addition to the individuals involved in animal farm activities, humans can be infected through unhygienic handling, processing, and/or consumption of fish, shellfish, and their products at the domestic, commercial, or household level [17,18,19,20,22]. Our study sequences are also found in the same cluster as human sequences. This result highlights the zoonotic potential of *V. cholera* that could be transmitted to humans directly through wild aquatic birds [23] or indirectly through environmental dissemination [24].

The investigation of the different geographical locations of the Nile revealed exposure of Nile water to several sources of municipal and industrial sewage pollution. Those pollution sources changed the characteristics of the Nile water and disseminated *V. cholerae* from other environmental niches. Interestingly, sediments have a polar electrical charge that is suitable for the colonization of *V. cholerae*. Sediments also constitute many zooplankton and phytoplankton that form micro- and macrohabitats, where *V. cholerae* can persist, multiply, and grow during a period of endemicity as the main reservoir for cholerae [4,11].

The epidemiological data have also illustrated the evident association between the population dynamics of *V. cholerae* and changes in physicochemical conditions and aquatic reservoirs affected by the spatial and temporal variations along the length of the Nile River in Giza, Cairo, and Delta provinces. In the natural environment, in addition to the availability of trace elements and chemical nutrients, several physicochemical parameters affect *V. cholerae* populations, including water temperature, sunlight, salinity, and pH [11,25,26,27].

Spatially, the characteristics of the Nile River ecosystem reflect the impact of the control of river water flow through its length that categorizes the Nile River into three zones: the High Dam lake in Aswan, Aswan to Nile Cairo, and the Nile Delta [13,14]. In natural ecosystems, salinity is one of the most important water parameters that can alter species distribution, life dynamics, biodiversity, and faunal composition, as different spatial characteristics are influenced by climatic, topographical, socioeconomic, and anthropogenic factors [28]. It has been found that the occurrence of cholerae is significantly related to increases in salinity. These data are in line with the work by Saad [29], who reported that the occurrence of cholerae in Egyptian marine aquatics was higher than in fresh water environments. The salinity of the Nile increased from 0.2 ppt in Aswan to 0.5 ppt in Cairo after the construction of the Aswan High and Old Dam, as a result of lowering the Nile water flow [30]. The salinity of Nile water in the Marriottya stream is 10 times higher than the Nile records in Aswan (0.2 ppt). This might be due to the dumping of several industrial and sewage wastewaters after the connection of the Marriottya stream to the Sakkara 7 drain [31]. Toward the Middle Delta, the salinity of the Nile water shifted suddenly to the brackish side, where a significant, noticeable occurrence of pathogenic *V. cholerae* in the finfish inhabits the mainstream of the Rosetta branch of Nile River at Kafr El-Zayat district, El- Gharbia province, Egypt. This shift in salinity is related to many topographical, climatic, and anthropogenic factors. In addition, the Rosetta branch of the Nile River at Kafr Elzayat city receives industrial effluents from Kafr El-Zayat Pesticides and Chemicals KZ, Salt, and Soda companies, which are discharged directly at the east bank of the branch [32]. From the far northeast to the northwest of the Delta, many authors have reported the marine shift in the salinity of the Nile water in Port Said, El-Behaira, and Alexandria provinces during the spring and summer seasons, respectively [13,33,34,35]. This salinity shift is a result of climate change, rising sea levels caused by global warming, and the intrusion of saltwater from the Mediterranean Sea downstream into the Nile.

With regard to temporal variation, the data presented here are in line with those of many authors, who have reported the occurrence of *V. cholerae* outbreaks in a stressful environment of high temperatures during the spring and summer seasons [13,26,27,28]. Islam [11] reported the synergistic association between the recurrence of cholera in Bangladesh and periods of high temperature with prolonged hours of sunshine, during which increasing photosynthesis and decreasing water volume resulted in a higher concentration of nutrients, green algal growth, and an increase in the aquatic bacterial load due to bacterial growth inside the algal sheaths. Another supportive study hypothesized that the onset of cholera epidemics after a period of endemicity is related environmentally to the decomposition of the algal bloom, which is also regulated climatically by the atmospheric water temperature and sunshine hours, in addition to the nutrient threshold conditions in the water [36]. Consequently, this scenario results in the release of bacteria into the aquatic environment in a condition associated with changes in reservoir growth and behavior and ultimately to the recovery, growth, proliferation, and sustainability of *V. cholerae* in the environment during the interepidemic periods.

About cholera dynamics, our data illustrated that *V. cholerae* inhabit different aquatic animals in different patterns as a result of relative host interactions among the examined species of cultured tilapia, catfish, or crustacean seafood. The disease terminology is also differentiated among the inhabited fish organisms. Some fish species are considered to be reservoirs, whereas others are potential vectors for cholera transmission. Although the examined samples were apparently healthy, some exhibited a clinical disease manifestation. In addition to environmental aspects, complex interactions related to species variability were revealed. As is known, fish act as reservoirs of *V. cholerae* [37], in which the bacterial form and number differ according to season, fish species, and natural habitat [38]. For *Tilapia niloticus*, our results agreed with those of Farouk [39], who reported a higher value of cholerae in examined sediment and water samples as compared with tilapia organs. However, at the time of harvest, fish carry a high microbial load on the surface of their skins, intestinal tracts, and gills, which might pose a public health risk after faulty handling, processing, or cooking by consumers, thereby acting as a potential vector. In addition, numerous species of waterfowl, such as cormorants, pelicans, seagulls, and ducks, usually feed on tilapia and other fish species from which *V. cholerae* have been identified [17].

In contrast, the occurrence of *V. cholerae* was significantly correlated with the presence of clinically diseased shield head fish (*Synodontis schall)* inhabiting the Rosetta Branch of the Nile River at the Kafr El-Zayat district. Shield head fish are bottom-feeding finfish that belong to the catfish species. They are widely distributed in North Africa and were discovered in Lake Nasser, Egypt [40]. Catfish species can survive in sewage, where the fish are exposed to high doses of untreated wastewater that includes free-living bacterial biofilms attached to several aquatic microfauna and flora. These findings were confirmed by Colwell [10], who reported similar environmental conditions helpful for *V. cholera* development and copepod attachment. The high water temperature and brackish shift of the Nile salinity in this district (site 3) might have caused an alteration in both the bacterial pattern and fish habitat, which resulted in infection and disease occurrence. This epidemiological interaction model supports the relationship between environment distress and the pathogenesis of *V. cholerae*, in which the roles of some pathogenicity factors do not have to be entirely related to the bacterial virulence. These perceptions assume that cholera is contracted by exposure to environmental reservoirs of toxigenic *V. cholera* that are directly driven by ecological factors.

With regard to crustacean fish, researchers have found a state of mutualism between the fish and the bacteria, in which *V. cholera* secrete many extracellular proteins as chitinase and protease, which in turn help the inhabited fish digest the proteins and chitin of the prey’s exoskeleton [16,39,41]. Therefore, it makes sense that crustacean seafood and copepods act as major reservoirs of *V. cholerae* within the aquatic environment. In addition, they are hunted by aquatic birds or unhygienically eaten raw by humans, resulting in the dissemination of cholerae to other environmental niches. The Nile Delta is part of the world’s most important migration route for birds. Each year, millions of birds pass between Europe and Africa along the East Africa Flyway, and the wetland areas of Egypt are especially critical stopover sites [35].

In the current study, we constructed a phylogenetic tree to explain the possible epidemiological relationship between the occurrence of cholerae in broiler farms and waterfowl (duck). We showed that the two *Omp W gene* sequences, broiler chicken (MT995936) and duck (MT 990929), which were isolated from poultry broiler and duck farms, were aligned with the other related *Omp W* gene sequences obtained from GenBank. The *Omp W* gene is a highly suitable genetic marker for comparing and sequencing *V. cholerae* strains because of its nucleotide sequence, which remained practically unchanged among strains isolated from the different sources [42]. The data reflect the potential role of aquatic birds in transmitting *V. cholera* to other poultry sectors. The exposure of the domesticated broilers is believed to have come from free-living waterfowl through fecal contamination of water sources [43]. This scenario is also supported by the phylogenetic tree analysis, as the studied duck sequence is more closely related to the lake water sequence (CP022352) than to the chicken sequence. In addition, diverse *V. cholerae* non-O1, non-O139 strains were recovered from an Austrian lake (Neusiedler See), and migrating birds were assumed to play an essential role in the transfer of such bacterial strains [44]. Other researchers have also reported that the migratory and residential waterfowl flourish on zooplanktonic copepods and chironomids, which can survive within the gut of water birds [45,46]. In addition, these viable zooplanktons can be attached to the external surface of waterfowl. These two findings indicate that waterfowl could potentially disseminate two main reservoirs of *V. cholerae*: fish species from which *V. cholerae* were identified and shellfish and crustaceans, from which waterfowl consume colonized planktons, fish, or crustaceans from different Nile districts.

Thus, it could be hypothesized that waterfowl serve as an intermediate vector for the cyclic transmission of *V. cholerae* between different environmental niches, including aquatic, poultry, human, and wildlife. The aquatic environment is a natural reservoir for cholerae, which is represented in all its natural components of water and sediment structure, algal blooms, water hyacinths, cyanobacteria, phytoplankton, zooplankton, and crustacean copepods. These changes in terminology might result in a better understanding of the ecoepidemiology of cholerae and provide an effective description of the occurrence of *V. cholerae*, which will finally break the cycle of transmission and prevent further cholera outbreaks.

In sum, studying the impact of the environment on models of the incidence cholera and different host interactions can provide a future fruitful pathway for cholera research.

## 4. Materials and Methods

### 4.1. Ethics Statement

In this study, we used freshly caught dead fish and poultry samples after routine fishing or slaughtering of birds at the end of the production cycle in different locations. All animal procedures were approved by the Institutional Animal Care and Use Committee, Faculty of Veterinary Medicine, Cairo University (VET. CU. IACUC; approval no. Vet CU16072020197).

### 4.2. Study Location and Duration

A surveillance study was conducted on the surface water bodies and fisheries located along the length of the Nile River in Giza, Cairo, to Delta provinces, Egypt, from which the different aquatic samples were collected during different seasons (from March 2019 through May 2020). In addition, poultry samples were collected from different poultry farms located within the same geographical zones in Giza and the Nile Delta, different districts from Port Said to Alexandria, approximately 1000 km along the middle coastal lines of the Mediterranean Sea (Figure 5).

Site 1 was located at the Marriottya canal, which is one of the intrastate branches of the River Nile, Egypt, extending through October, and Giza governorates, a 4-km distance that ranges from Shabramant to Abouseer city (Figure 5, site 1).

Site 2 was the Nile in Cairo at the Al-Maadi district, which is parallel to the Corniche, a waterfront promenade, and the main road north into Cairo. Water, sediment, and finfish samples were collected from the Nile near the National Egyptian Geological Museum (Figure 5, site 2).

The Nile Delta is one of the largest river deltas in the world. It is formed in Lower Egypt, where the Nile River extends and drains into the Mediterranean Sea. After the construction of the Aswan Dam, the Nile Delta stopped flooding annually. The Delta is divided into different geographic districts by the main of the two Nile distributaries: the Damietta and the Rosetta branches. It extends along the coastal line of the Mediterranean Sea from Alexandria West to Port Said East [35]. The Rosetta Nile branch is impacted environmentally by various concentrations of wastewater from different pollution sources.

At site 3, at Kafr El-Zayat city, El-Gharbia province, Egypt, the Nile water in the Rosetta branch is influenced by the activities of industrial companies as well as agricultural drains located along its sides. It receives industrial discharges from many companies that drain directly at the east side of the branch. On the other side, Nile water receives agricultural and domestic drainage wastewater through EL-Rahawy, Sabal Drain, El-Tahreer Drain, Zaweit El-Bahr Drain, and Tala drains. Topographically, there was a stake zone of brackish and marine water extending into the aquifer in the Middle Delta to 90 km from the Mediterranean Sea coast [47]. Most of the freshwater that reaches the Nile Delta is diverted and channeled into complex networks for the distribution of agricultural water. These open and inefficient irrigation canals result in a high rate of evaporation of the Nile’s freshwater [48] (Figure 5, site 3).

Similar to other Delta districts, the Nile Delta is subject to the adverse impact of global warming and climate change, which has resulted in Delta subsidence, erosion, accretion, and sea-level rise [49]. Moreover, Egypt is predicted to be one of the top five countries severely affected by sea-level rise. Alexandria, El-Behaira, Port Said, and Damietta Governorates have been identified as high-risk vulnerable areas of the Nile Delta with possible socioeconomic impacts [35]. Toward the Northeast Delta, the Northern Lakes represent the final reservoirs and chance for Nile water usage before it drains into the sea.

Site 4 was located at West Port Said, where the Manzala Lake links the Mediterranean Sea through Boughaz El-Gamil (Figure 1, site 4). In addition, this site receives drainage water through the five main drains of Bahr-Bagar, Al-Gamaliah, Al-Serw, Bahr Hadus, and Ramsis [35].

Sites 5 and 6 comprise the northwest to the Rosetta branch, where the El-Mahmoudeya canal runs about 77 km to meet the freshwater drinking and irrigation demands of several municipal, agricultural, and industrial activities as well as fisheries in El-Behaira (Figure 5, site 5) and Alexandria provinces, Egypt (Figure 5, site 6). In the Rosetta Estuary, the freshwater of the Nile ends 30 km upstream of the sea, where it releases excess water into the Mediterranean Sea [50,51].

### 4.3. Collection of Different Samples

Different samples were collected from aquatic environments (water, sediment, finfish, and crustacean seafood samples), which is in line with surveys of different poultry farms (broiler chicken, duck, and turkey) located within the same geographical zone.

#### 4.3.1. Aquatic Sources (Water, Sediment, Finfish, and Crustacean Seafood)

Water samples and sediments

Water samples of 250 mL each were collected from the midstream and sediment of the Nile River surface bodies and fisheries in Marriottya, Maadi, and Kafr El-Zayat (Middle Delta) districts in addition to the coastline of the Mediterranean Sea (Figure 5). Two samples were collected from each site. Water temperature, pH, and salinity were determined in situ using a digital thermometer (0 °C–100 °C; HANNA^®^), pH meter (Jenway^®^), and a refractometer (portable optical TDS salinometer/refractometer), respectively (Figure 1). All water and sediment samples were collected into sterile 500-mL glass jars, labeled carefully, and transported directly inside an icebox surrounded with ice gel packs to the research laboratory of the Department of Veterinary Hygiene and Management, Faculty of Veterinary Medicine, Cairo University. To isolate *V. cholerae* from water samples, we filtered the 10 water samples using a membrane filtration system unit (BiomMerieux^®^) into a sterile filter membrane (0.45 µm) that retained bacteria. Subsequently, we used sterile forceps to transfer the filter membrane into alkaline peptone water (APW) for selective enrichment after appropriate mixing using the vortex (BRP^®^, H05W-F) [52]. Sediment samples were enriched directly with double-volume sterile APW without a filtration step.

Finfish samples

A total of 25 finfish of Nile tilapia (*Oreochromis niloticus*) and shield head fish (*Synodontis schall*) were collected from the Nile River from the prementioned districts in Giza, Cairo, and Al-Gharbia provinces, Egypt (Figure 5, sites 1, 2, and 3) together with sediment and water samples. Freshly caught fish samples were placed in double sterile polyethylene bags and transported inside a dry, clean icebox to the laboratory for clinical and postmortem examination and bacterial isolation of *V. cholerae.* All collected fish samples were examined clinically according to the methods of Amlacher [53] for the presence of abnormal signs in the tail, fins, eyes, and external slime layers. Then, fish were sectioned for postmortem examination of color and volume of abdominal fluids in addition to abnormal lesions in the internal organs. Finally, a collection of different specimens from the gills, liver, kidney, and intestine using sterile bacteriological swabs was transferred into APW for enrichment (Table 3).

Crustacean fish (seafood)

A total of 75 Kuruma prawn shrimp samples (*Penaeus japonicas)* were collected from different districts located along the coastal line of the Mediterranean Sea in Port Said, El-Behaira, and Alexandria provinces (Figure 5, sites 4, 5, and 6). We pooled the shrimp samples according to the American Public Health Association (APHA) [54]. Groups of three to five small shrimps were pooled and homogenized using 10-mL sterile APW and a Stomacher^®^, but large-sized samples were divided into three body specimens: the cephalothorax, abdomen, and tail (Table 3).

#### 4.3.2. Poultry Sources

A total of 115 samples of intestinal content were collected from 12 broilers, 7 duck, and 4 turkey commercial poultry farms, feed on Nile River dikes, or located close to the examined Nile districts (Figure 5). The intestines were collected just after the slaughter of different poultry species in sterile jars containing 500 mL of sterile saline solution, and 10 mL was transferred into APW for the enrichment of *V. cholerae.*

### 4.4. Phenotypic Identification of V. cholerae (Isolation, Culturing, and Biochemical Characterization)

The presumptive identification of *V. cholerae* was carried out following the Bacteriological Analytical Manual procedures of vibrio (BAM: vibrio) [55]. All samples were enriched using APW broth at 35 °C ± 2 °C for only 6 to 8 h. As a result of the rapid generation time of *V. cholerae,* short incubation periods are the best for isolation and to minimize other competing microflora that might overgrow *V. cholerae* after a long incubation period. Culturing proceeded in the same incubation conditions, where a loopful culture from enriched APW was streaked onto the surfaces of dried thiosulfate citrate bile salt agar plates (Oxide^®^). Then, suspected colonies were picked and subcultured into tryptic soya agar supplemented with sodium chloride 2% agar plates (TSA-2% NaCl) for purification at 35 °C ± 2 °C for 6 to 8 h.

Then, we conducted a biochemical characterization of the suspected colonies, in which we tested the following: oxidase reaction, salt tolerance on tryptone–NaCl 0% (T1N0), and NaCl 3% (T1N3), in addition to gelatin and arginine hydrolysis using arginine glucose slant and nutrient gelatin slants (Oxide^®^), respectively. We also detected the activity of both ornithine decarboxylase and urease.

### 4.5. Molecular Identification of V. Cholerae

We extracted DNA from the isolates using the boiling method. In brief, a loopful of cultured bacteria was suspended in 100 μL of sterile Tris-EDTA (TE) buffer. The samples were boiled for 10 min at 100 °C and then immediately cooled in ice water. Samples were centrifuged for 5 min at 6000× *g* to remove the supernatant. The extracted DNA was stored at −20 °C until use [56]. Routine PCR was adopted for molecular detection of the universal 16S rRNA gene and specific target outer membrane protein *(Omp W*) of cholera strains. PCR amplification was performed for 5 min at 94 °C in a total reaction volume of 25 μL containing 12.5 μL PCR master mix (Sigma), 4.5 μL PCR-grade water, 1 μM of each primer (10 pmol), and 6 μL DNA template. For the initial denaturation of DNA in 35 cycles, each cycle lasted 30 s at 94 °C, 2 min at 59 °C, and 1 min at 72 °C, with a final extension for 7 min at 72 °C in a DNA gradient temperature cycler. For each reaction mixture, 10 μL was subjected to electrophoresis on a 1.5% agarose gel. The gel containing the amplified DNA was stained with ethidium bromide and visualized with an ultraviolet transilluminator. Then, images from the transilluminator were digitized with a one-dimensional gel documentation system (BioRad).

### 4.6. Sequence Analysis

The *Omp* W gene from *V. cholerae* isolates was randomly selected from broiler chicken and duck samples from the same locality. Then, amplified fragments were purified using the QIA quick Gel Extraction Kit (QIAGEN) according to the manufacturer’s instructions and sequenced at Promega Lab Technology using the forward and reverse primers of the *Omp W* gene listed in Table 5. The gene sequences were deposited in the National Center for Biotechnology Information (NCBI) GenBank database under the accession numbers MT 995936 and MT 990929 for the chicken and duck-derived sequences, respectively. The nucleotide sequences from the current study were compared with fifteen sequences available in the NCBI GenBank databases using the NCBI BLAST server. Then, we performed phylogenetic tree analysis with MEGA version 7 by using a neighbor-joining approach with 1000 bootstrapped replicates.

### 4.7. Statistical Analysis

We performed analysis of the data using PASW Statistics version 18.0 software (SPSS Inc., Chicago, IL, USA). The occurrence of *V. cholerae* in the Nile River with regard to different environment and host factors was determined by chi-square (χ^2^) or Fisher’s exact tests. A *p* value ≤ 0.05 was statistically significant.

## 5. Conclusions

*V. cholerae* inhabits different Nile environments, where various cholera models were found to be related to the variability of the species of the examined aquatic samples of both finfish and crustacean seafood, in addition to the temporal and spatial variation of their environments. We found that waterfowl were potential vectors for the intermediate transmission of *V. cholerae* between the different environmental niches. Therefore, it is essential to continuously monitor the quality of Nile water alongside any water quality alterations to prevent the outbreak of health disorders as well as to mitigate river pollution, as the impact of this pollution has become significant during low Nile flow years. In addition, to break the cyclic transmission between human, aquatic, and animal sectors, it is imperative to follow good management practices and biosecurity rules in the field of animal production in addition to following general sanitary practices and hygienic guidelines, most strictly in the handling, processing, and consumption of animals and animal products.

## Figures and Tables

**Figure 1 pathogens-10-00190-f001:**
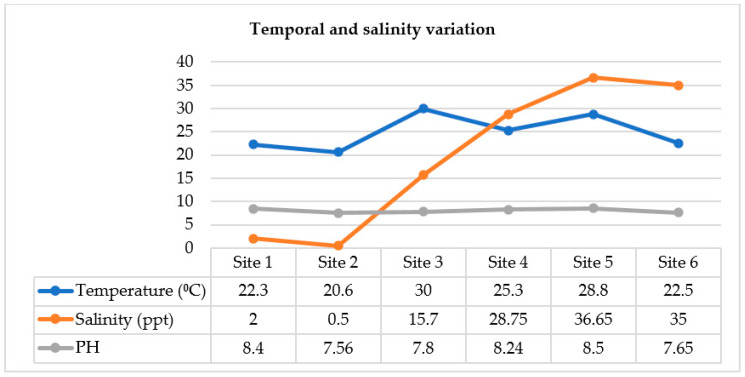
Physiochemical conditions of examined aquatic Nile sources.

**Figure 2 pathogens-10-00190-f002:**
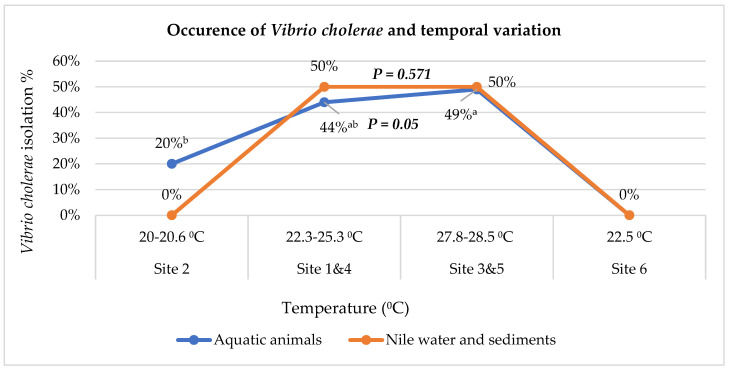
Occurrence of *Vibrio cholerae* in the Nile environment, examined finfish, and crustacean seafood aquatic sources regarding temporal variation. ^a,b^ Different superscripts indicate significance at *p* ≤ 0.05.

**Figure 3 pathogens-10-00190-f003:**
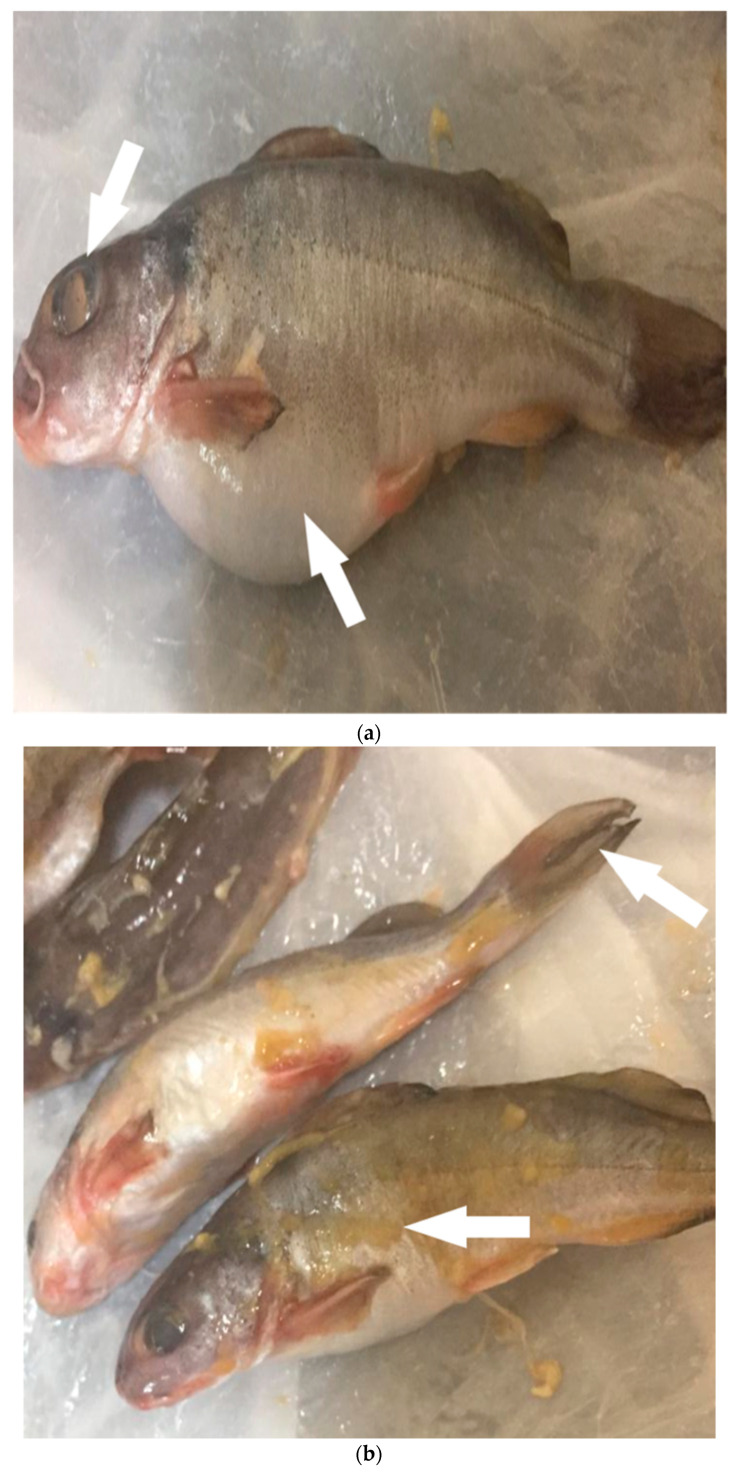
(**a**) Shield head fish (*Synodontis schall*) with symptoms of abdominal dropsy and corneal opacity. (**b**) Shield head fish with symptoms of skin discoloration, fin rot, and scale detachment.

**Figure 4 pathogens-10-00190-f004:**
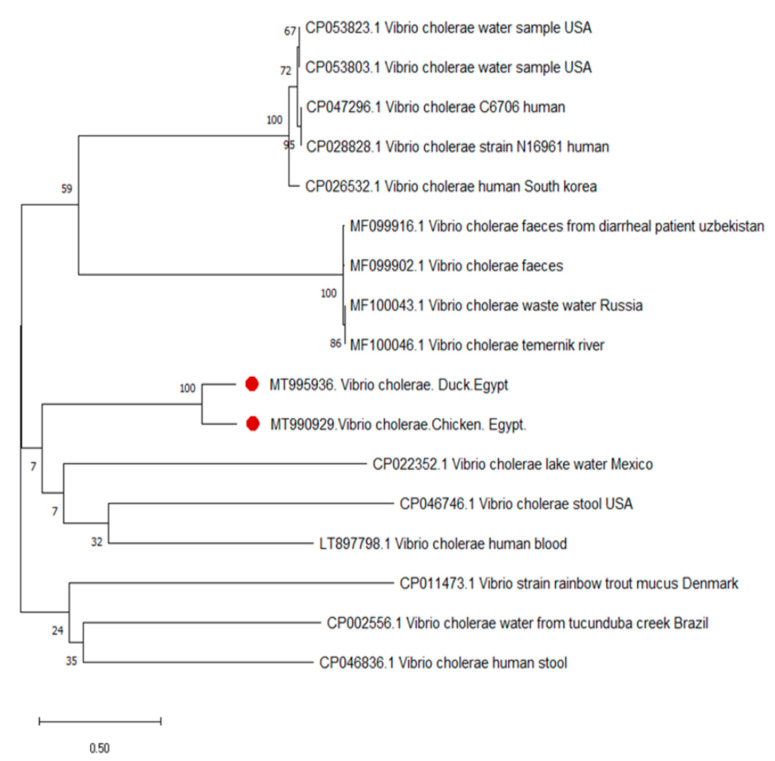
Neighbor-joining tree based on the *Omp W* gene sequences showing the relationship among our study sequences and 15 representative sequences retrieved from GenBank. The studied sequences are indicated by bullets. The evolutionary analysis was performed using MEGA version 7.

**Figure 5 pathogens-10-00190-f005:**
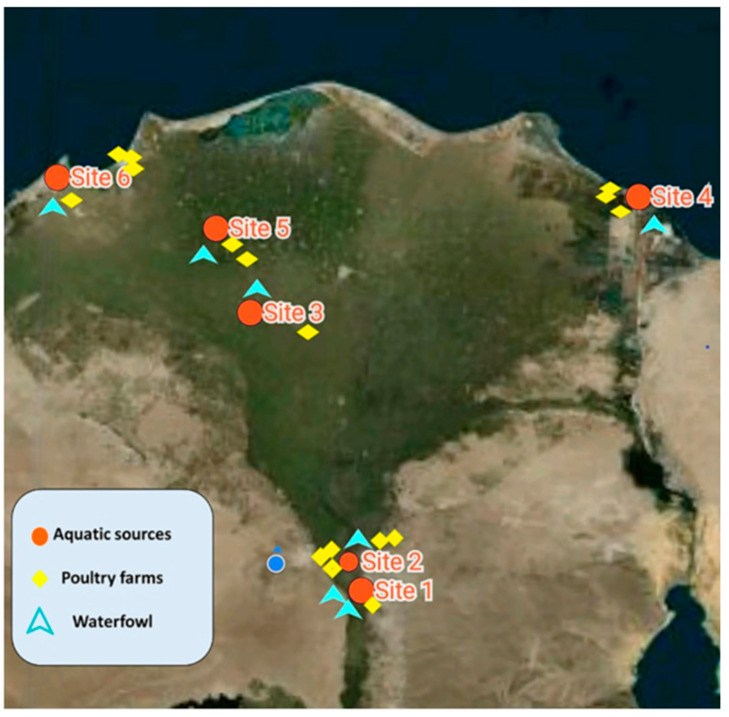
Site selection and sampling points of aquatic environments, waterfowl, and poultry.

**Table 1 pathogens-10-00190-t001:** Occurrence of cholera in different environmental niches.

Environmental Niche	No. of Examined Samples	*V. cholerae* Isolates (%)
Finfish and crustacean seafood	185	83 (45%)
Water and sediment	12	5 (42%) (4 sediments and 1 water)
Waterfowl, broiler, and turkey	115	41 (36%)
Total	312	129/312 (41%)

**Table 2 pathogens-10-00190-t002:** Occurrence of *V. cholerae* regarding salinity variations of the Nile aquatic environment.

Nile Salinity	Location	*V. cholerae* Isolates (%)	
		Water and Sediments	Fish Sources
Fresh water	Sites 1 and 2	25%	33.3% ^b^
Brackish water	Site 3	50%	71.9% ^a^
Marine water	Sites 4, 5, and 6	50%	29.4% ^b^
0.773 (FET)		<0.0001	

^a, b^ Different superscripts indicate significance at *p* ≤ 0.05. FET, Fisher’s exact test.

**Table 3 pathogens-10-00190-t003:** Occurrence of *Vibrio cholerae* in fin fish and crustacean seafood aquatic sources.

Environmental Location	No. of Examined Samples	Specimens	No. of *V. cholerae–* Positive Samples/Pools	Total No. of *V. cholerae* Isolates
South Delta Site 1 Site 2	9 Nile tilapia (*Oreochromis niloticus*)	Gills	3	12/36 ^b^ (33.3%)
		Kidney	3	
		Liver	3	
		Intestine	3	
Middle delta Site 3	16 Shield head fish, catfish (*Synodontis schall*)	Gills	16	46/64 ^a^ (72%)
		Kidney	12	
		Liver	12	
		Intestine	6	
North Delta Site 4 Site 5 Site 6	75 Shrimp *Penaeus japonicas*	14 pools	4 pools (each 5 shrimps) = 20 shrimps	25/85 ^b^ (29.4%)
		5 Cephalothorax	3	
		5 Abdomen	0	
		5 Tail	2	
Total aquatic samples	100 sample	185	83	83/185 (45%)
*p* value				0.0001

^a, b^ Different superscripts indicate significance at *p* ≤ 0.05.

**Table 4 pathogens-10-00190-t004:** Occurrence of cholera in different poultry species.

Poultry Species	No. of Examined Samples	*V. cholerae* Isolates (%)
Duck	35	14 (40%) ^a^
Broiler chicken	60	27 (45%) ^a^
Turkey	20	0 ^b^
Total	115	41 (36%)
P value		0.001

^a, b^ Different superscripts indicate significance at *p* ≤ 0.05.

**Table 5 pathogens-10-00190-t005:** Primer nucleotide sequence for *16S*
*rRNA* and *Omp W* genes.

Name of Amplified Region	Primer	Sequence	Length of Amplified Product (bp)
*16S Rrna*	F R	5′-CAGGCCTAACACATGCAAGTC-3′ 5′-GCATCTGAGTGTCAGTATCTGTCC-3′ [57]	700 bp
*Omp W*	F R	5′-CACCAAGAAGGTGACTTTATTGTG-3′ 5′-GAACTTATAACCACCCGCG-3′ [43]	587 bp

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
