# Peer review of "Ecoepidemiology and Potential Transmission of Vibrio cholerae among Different Environmental Niches: An Upcoming Threat in Egypt"

_pathogens, 2021, doi:10.3390/pathogens10020190_

Round 1

Reviewer 1 Report

The paper pathogens-1081356 takes into account a real health emergency in many countries, including Egypt where the present study is located, such as cholera.

The manuscript is well coordinated in the dynamics of sampling and in the provision of related data, considering that several places along the Nile River shaft are monitored, up to a delta area.

Furthermore, more possible sources of bacterial pollution are considered in addition to water, such as fish fauna and birds.

The introduction is well conducted and sufficiently clear; the materials and methods are well characterized and the monitoring sites, the sampling methods of the various possible sources of contamination and the identification of the causative agent are well described.

As for the results, we must keep the data entered good, even if it is not a similarity between the samplings made in the different matrices, in order to have a real correlation between the various data obtained. Despite this gap, the results obtained support the discussion carried out by the authors which appears well argued and clear.

As regards specifically the results obtained on the assessment of occurrence with respect to seasonality, they cannot be clearly linked as the sampling was not conducted in all seasons for all sites.

Therefore, the results entered are only a snapshot of the time situation, not justifiable for the objectives set. In line 138 the inserted site (site 3) does not correspond to what is shown in the graph of figure 2 (where the negativity to cholera is in site 2).

So this chapter must certainly be reviewed in light of the paucity of the reported results. Even in the following chapter, the examination of aquatic organisms raises some perplexity as it was set up; first of all, in order to be able to correlate the data in a useful way, it is necessary to compare an equal number of subjects (perhaps a little higher than the one implemented). 

Furthermore, for the evaluation of the pathogenic potential of the germ on fish, the gill and intestine matrices are of little significance. Then the data should be taken, cleaned of the listed biases and re-evaluated. The photos inserted are unclear and deformed: you need to insert images of better quality and of the correct format so as not to deform the image itself.

That said, the value of the work is good and undoubtedly interesting for publication, even if there are some topics to improve and correct.

Based on this it is my opinion that the work should be revised with major revision.

Author Response

Dear Reviewer,

Alot of thanks and appreciation for you kind fruitful and supportive reviewing, I hope my response meet your acceptance.

Reviewer 2 Report

To the authors:

Although Vibrio cholerae is a major threat of global health, our knowledge about the life cycle of this bacterium und and its occurrence in the environment is still very limited. In this study, Ismail et al. isolate V. cholerae from various environmental sources associated with the Nil Delta and study the abundance of this bacterium in various hosts and seasonal conditions. This study is of interest to the field of V. cholerae research as it (i) presents novel isolates and (ii) provides genotypic and phenotypic data of these isolates and (iii) correlates the abundance of V. cholerae with the season and the host.

Major comments:

l.101ff The text refers to the isolates being “tested positive for oxidase, ornithine decarboxylase, and gelatinase activity, whereas they 105 tested negative for arginine hydrolysis and urease activity.” I don’t think any data is provided to support these statements. I suggest adding the corresponding data to the manuscript or at least to the supplement.

Figure 2, I recommend labelling the y-axis. Maybe I missed this, is the total number of samples taken at each timepoint indicated somewhere so that it is clear what 100% refers to?

Figure 4, This tree gives an idea of the relatedness of the isolates of this study to isolates of other studies, which is nice. I would recommend to also include the ompW sequences of the strains N16961 and C6706. Both of those strains are heavily studied in labs and the relatedness of those two strains to the isolates of this study might also be of interest to the readers.

Minor comments:

Table1, I don’t see the need for statistics here.

l. 29, no need to indicate a p-value in the abstract already

Author Response

Dear reviewer,

I would like to express my great thanks and appreciation for your kind, fruitful, and supportive scientific input of our study.

Reviewer 3 Report

Dear authors.

The manuscript submited to Pathogens is suitable for publication after major revision.

Although the monitoring of V. cholerae in different Nile environments is very interesting, the methodology employed to explain the possible epidemiological relationships between different isolates is not adequate.

I consider that to perform an epidemiological study, another techniques as MLST, MLSA which implicate the study of a major number of genes highly conserved, should be employed. The authors analyze only the Omp W gene sequences of two isolates and compare with other available in the NCBI databases. For me this is not enough.

Which is the “outgroup” in the phylogenetic tree? It is possible that if in the phylogenetic analysis is included an appropiate specie as the “outgroup”, all the V.cholerae strains employed constitute an unique branch in the tree.

In basis to the methodology and the results obtained, I consider that its is impossible to stablish the transmission cycle of V.cholerae among different niches.

Table 5 is not neccesary. The primers sequences could be included in the materials and methods section.

Author Response

Dear reviewer, 

I would like to express my great thanks and appreciation for your kind, supportive, and fruitful scientific input of our study.

I am sorry that you kindly recommend us to use other techniques as MLST, MLSA in this study, there was a great limitation. MLST and MLSA are not available or applicable in Egypt besides other difficulties limited this point

I hope my response and explanations meet your acceptance.

Round 2

Reviewer 1 Report

The authors of the pathogens_1081356 manuscript have sufficiently integrated the text with the requests made or justified the choices made. The authors of the pathogens_ manuscript have sufficiently integrated the text with the requests made or justified the choices made.

There is still a problem with the photographs used and precisely in figure 3 the fish is altered in size: it should be resized as the subject appears deformed and does not correspond to the real size (it is wider than long ... see the next figure to understand the dimensional relationships)There is still a problem with the photographs used and precisely in figure 3 the fish is altered in size: it should be resized as the subject appears deformed and does not correspond to the real size (it is wider than long ... see the next figure to understand the dimensional relationships)

Author Response

Dear reviewer thank you again for your kind, fruitful, and continues review of our manuscript

I will ask for help toward fig. 3, otherwise, I will remove them with their citation from the data section. I have tried to improve them to the required dimensions, but it seems that I failed. I am so sorry about this.

Reviewer 3 Report

Dear authors.

In base to the response and the changes performed in the manuscript , I consider it as accept.

Author Response

Dear reviewer: Great thanks and appreciation for your kind, helpful, and fruitful comments.

This manuscript is a resubmission of an earlier submission. The following is a list of the peer review reports and author responses from that submission.

Round 1

Reviewer 1 Report

The article pathogens_1045163 takes into consideration with an eco-epidemiological approach a very important topic for the risk on public health, such as cholera.

The development of the topic is undoubtedly multidisciplinary and touches on various environmental aspects, taking into consideration biotic and abiotic factors of the terminal part of the Nile and its delta.

Having analyzed both water and sediment and various potential carriers of the pathogen of both fish and avian nature is undoubtedly a strong point.

I would have liked to have been able to evaluate a much higher numerical consistency for all types of samples examined to have more robust data.

In my opinion it remains to evaluate and better highlight the results obtained in the various matrices, in the different sampling periods, always considering all the sampling sites.

The introduction is well conducted by the authors. In the results the photos of the fish examined are not clear; it would be necessary to improve its quality.

The discussion is extensive and well documented, as well as the materials and methods are sufficiently described. The description of the sampling sites is well organized. In the conclusion it would be good to emphasize that this study has a preliminary approach and that the results will have to be re-evaluated with a more numerous and well structured sampling.